# Epidemic and Pandemic Preparedness and Response in a Multi-Hazard Context: COVID-19 Pandemic as a Point of Reference

**DOI:** 10.3390/ijerph21091238

**Published:** 2024-09-19

**Authors:** Thushara Kamalrathne, Dilanthi Amaratunga, Richard Haigh, Lahiru Kodituwakku, Chintha Rupasinghe

**Affiliations:** 1Global Disaster Resilience Centre, School of Applied Sciences, University of Huddersfield, Queensgate, Huddersfield HD1 3DH, UK; dr.d.amaratunga@gmail.com (D.A.); r.haigh@hud.ac.uk (R.H.); 2Ministry of Health, Colombo 01000, Sri Lanka; lahirundcu@gmail.com (L.K.); suranjalee@yahoo.com (C.R.)

**Keywords:** epidemics, pandemics, COVID-19, pandemic preparedness, pandemic response, multi-hazard preparedness

## Abstract

Infectious diseases manifesting in the form of epidemics or pandemics do not only cause devastating impacts on public health systems but also disrupt the functioning of the socio-economic structure. Further, risks associated with pandemics and epidemics become exacerbated with coincident compound hazards. This study aims to develop a framework that captures key elements and components of epidemic and pandemic preparedness and response systems, focusing on a multi-hazard context. A systematic literature review was used to collect data through peer-reviewed journal articles using three electronic databases, and 17 experts were involved in the validation. Epidemiological surveillance and early detection, risk and vulnerability assessments, preparedness, prediction and decision making, alerts and early warning, preventive strategies, control and mitigation, response, and elimination were identified as key elements associated with epidemic and pandemic preparedness and response systems in a multi-hazard context. All elements appear integrated within three interventional phases: upstream, interface, and downstream. A holistic approach focusing on all interventional phases is required for preparedness and response to pandemics and epidemics to counter their cascading and systemic effects. Further, a paradigm shift in the preparedness for multi-hazards during an epidemic or pandemic is essential due to the multiple challenges posed by concurrent hazards.

## 1. Introduction

COVID-19 was declared a Public Health Emergency of International Concern (PHEIC) by the World Health Organization (WHO) on 30 January 2020 and a global pandemic on 11 March 2020. It followed the dramatic increase in infection and death rates resulting from the severe acute respiratory syndrome coronavirus 2 (SARS-CoV-2), which was initially found in China’s Hubei province in early December 2019 [1,2]. There has been a wide variation between countries regarding the effectiveness of preparedness and response strategies in combating unpredicted constraints and challenges encountered during the COVID-19 pandemic [3,4]. A few countries, which had previously experienced large-scale infectious diseases, and countries with improved risk governance appeared more able to lower the impact of the pandemic outbreak [5,6]. However, many developing nations were severely impacted due to a lack of preparedness, ineffective strategies in mitigation and response, and other primary socio-economic factors [7]. The COVID-19 pandemic demonstrated that no nation could survive independently during a global public health emergency, stressing the need to build resilience against global public health challenges and strengthen collective efforts to improve global preparedness and response.

Even though devastating pandemics and epidemics are not new, the far-reaching socio-economic consequences of the COVID-19 pandemic are unparalleled compared to previous disease outbreaks [8]. Recent human civilization has been through several pandemics and epidemics caused by infectious diseases that have repeatedly appeared [9,10]. Therefore, lessons learnt from past pandemics and epidemics should inform mitigating the risk of present and future pandemic events, as many diseases can re-emerge or be re-introduced [11]. The systematic nature of a pandemic’s impact reaches beyond the health sector. As evidenced by COVID-19, a global pandemic has the potential to undermine economic and political stability and disrupt social order and integrity [12]. Such a threat requires the deployment of effective preparedness and response strategies to build global resilience in the public health sector. International health actors have identified the concept of global health security for governing relevant national and global agencies, including national governments, as a collaborative body to streamline policies and activities in pandemic preparedness and responses [13].

Notably, the risks and vulnerability of health hazards in some countries have been exacerbated by multi-hazard scenarios, including concurrent events such as those triggered by meteorological and seismic hazards [14,15]. Some nations experienced unprecedented challenges in their public health sector because of concurrent disease outbreaks, for example the dengue outbreak in Sri Lanka and Ebola in Congo [16,17]. Considering the potential for concurrent hazards and cascading impacts, strengthening multi-hazard preparedness and response has been widely recognized as a global priority [1,18].

COVID-19 is a crucial landmark for empirical investigations to understand the critical components of pandemic warnings, preparedness, and response [19]. Critical elements of preparedness and response systems have been identified during previous epidemics and pandemics that help in exploring some generic elements common to many infectious disease outbreaks [10,13]. Also, several studies conducted during COVID-19 to understand the essential variables of the global pandemic have significantly contributed to identifying essential components linked with pandemic preparedness from a systemic perspective [20,21,22,23,24]. However, many of the current frameworks and preparedness strategies have focused primarily on a public health emergency perspective, and thus a compound hazard perspective has been largely overlooked [25,26]. Therefore, studies need to investigate how pandemic preparedness systems could be better incorporated with complex situations, such as concurrent natural hazards amidst disease outbreaks [27,28,29]. Considering this gap, this study intends to identify key elements and components of pandemic preparedness and response focusing on a multi-hazard ecosystem, which helps to understand what components need to be integrated within a complex disaster preparedness system to address complex challenges encountered during a combined or compound hazard scenarios [14,18,30,31]. Also, new insights and approaches are required to understand complex connections in emergency management, as biological and natural hazards do not fit well within the same typical emergency management structures [32]. Further, preparedness and response to other hazards are more complicated when concurrent health hazards produce a double burden [30,33]. Notably, many developing countries are far behind in integrating biological hazards into their emergency management and disaster risk reduction (DRR) instruments, as national frameworks are largely natural hazard-oriented [34,35]. Within this context, this paper is an account of a study to identify and analyze the key elements and components of epidemic and pandemic preparedness and response, which aimed at developing a conceptual framework that provides a fully integrated system for epidemic and pandemic preparedness and response in a multi-hazard context. The proposed framework focused on COVID-19 to capture the relevant components and elements, fostering its broader use in future infectious diseases with epidemic or pandemic potential.

## 2. Materials and Methods

This section describes the methods and procedures adopted for the desk study, including the literature review protocol and validation of the findings carried out in Sri Lanka.

In the first stage, a desk study was conducted to understand the background of the COVID-19 pandemic, previous epidemics and pandemics, interventions used by different countries during infectious disease outbreaks, and the role of international and domestic health and DRR authorities in COVID-19 prevention and control to scope the research. The literature review focused on overall aspect of the preparedness and response strategies to be applied during an epidemic or pandemic. The systematic review was used to capture the main elements and components of a pandemic preparedness and response process, focusing on a multi-hazard context to develop a conceptual framework. The following guiding questions were used to help scope the systematic review.

What are the key components of epidemic and pandemic preparedness and response systems?What are the key phases of a pandemic and/or epidemic?How can epidemic and pandemic preparedness be integrated into a multi-hazard strategy?

The second stage dealt with validating the conceptual framework using the views of experts in the relevant fields, as described in Section 2.2 and Section 5.

### 2.1. Methods Used in Literature Mining

A systematic literature review was used to gather data on key components and phases of epidemic and pandemic preparedness and multi-hazard preparedness strategies. The systematic review method increases transparency and reproducibility, and a high level of subjectivity is encountered in data collection using traditional, non-structured literature reviews [36]. High-quality systematic literature reviews support better decisions for policymaking by systematically reviewing published research and synthesizing the results with a high level of objectivity [37]. Accordingly, this research administrated procedures and protocols of systematic literature review methods, described as follows.

The initial article search was conducted to formulate an overview of the literature published on the COVID-19 pandemic and other epidemics and pandemics reported in recent history. The article search was undertaken carefully, using a combination of appropriate search strings to collect a controllable number of empirically robust and comprehensive papers. The article search was initiated using three electronic databases, PubMed, Scopus, and ScienceDirect, at three independent instances. The literature searches and reviews were administrated entirely based on the broader research question of what the key components and phases of epidemic and pandemic preparedness and response ecosystem are.

The literature mining process started once the keywords were derived from the research question. A Boolean search operator was initiated using the following syntax, formulated by combining selected keywords: COVID-19 OR Pandemic OR Epidemic AND components OR Phases AND Response OR Preparedness. Literature mining was carried out from 5 to 12 February 2022.

Thirty-two (32) research papers were selected for analysis after a carefully conducted literature mining process. Literature mining executed using electronic databases was restricted only to published peer-reviewed research articles and disregarded unpublished works, working papers, abstracts, books, chapters, thesis, and meta-analyses to limit the research articles to a manageable amount for the study purpose and to filter high-quality research articles for further use. The search was limited to papers published between 2010 and 2022 to understand the trends and patterns of recent epidemics and pandemics reported during the previous decade. In addition, areas were limited to public health, epidemiology, social sciences, environmental sciences, computer science and information management, and multidisciplinary research.

The review adopted a standard literature mining and filtering procedure. Using PRISMA, the following flow chart elaborates on each stage (Figure 1) of the literature search and manuscript selection process. The following inclusion criteria were applied in the selection of manuscripts.

The main focus is primarily on the preparedness and response for epidemics and pandemics, biological hazards, or infectious diseasesContaining approaches, strategies, or mechanisms in managing public health emergencies or multi-hazardsThe perspective of the manuscript is disaster risk reduction or emergency managementIncluding elements, components, stages, phases, or frameworks in epidemic or pandemic preparedness and response

The exclusion criteria used in the manuscript selection were:The main perspective of the paper is purely medical and thus does not reflect the broader interdisciplinary aspect of the preparedness and response during an epidemic or pandemicThe manuscript contains only statistical or meta-analysis that cannot be used to grasp any critical elements or components of pandemic preparedness

Out of 12 potential records, 6 grey literature sources were included in the literature review, considering their relevance to the broader themes of the research questions. Most grey literature sources have been published by the WHO, UNDRR, The World Bank, and the European Union as technical reports. In addition, a few journal papers were selected purposely to reflect definitions of concepts used in the analysis. Overall, grey literature sources were chosen purposefully, considering their rigor in explaining pandemic preparedness measures and interventions. The selected reports have been published on COVID-19 preparedness and response, managing risks in epidemics and pandemics, infectious disease preparedness and response interventions, and multi-hazards faced by different countries during the COVID-19 pandemic. Except for the World Bank report on multi-hazard response interventions during COVID-19, which is one of the key focuses of this study, all other reports have been published by mandated international health and DRR authorities and highlight epidemic and pandemic preparedness and response interventions that have taken place amidst the COVID-19 pandemic. Therefore, grey literature sources were carefully selected to understand key components and phases of epidemics and pandemics in terms of preparedness and response strategies and interventions implemented by global health authorities.

Altogether, 38 literature sources were selected for the analysis. Once the selection was finalized after the full paper screening, all research articles were imported into the citation manager (EndNote X9). Data were analyzed using qualitative techniques, primarily thematic and synthesis methods. First, the descriptive themes were extracted from the selected literature sources, and then analytical themes were identified. The synthesizing method was used to establish the interconnectedness between the analytical themes embodied in the conceptual framework.

### 2.2. Validation Methods and Procedures

The framework was validated in Sri Lanka in relation to the country’s public health emergency preparedness and response system to assess the validity of the identified elements and phases.

Seventeen experts in relevant fields were involved in the validation, which was driven through four rounds of consultative meetings. They were from DRR and health emergencies, health experts attached to the Ministry of Health, universities, disaster management centers, and local authorities. They provided a strong partner and stakeholder network, which enabled the framework to be validated empirically at the national level.

Consultative meetings and discussions have been identified as a rigorous strategy to ascertain expert opinions in validating frameworks [38,39]. Ng’etich et al. [39] have identified two methods of expert validation of a framework initially developed by a desk study: (i) consultative meetings with experts and stakeholders and (ii) presentation of the draft framework in scientific sessions constituted of researchers and policy experts. Similarly, ascertaining expert opinions and presenting a preliminary framework in an expert session to obtain inputs have been used by Calciolari et al. [40] for validating a draft conceptual framework. Accordingly, the framework developed in this study also used expert consultative meetings and presenting preliminary frameworks in scientific sessions as two potential validation strategies.

The first three rounds of consultative meetings were organized as hybrid meetings in which Sri Lanka-based experts participated physically and members who joined from abroad attended online. The final validation round was organized as a workshop in Sri Lanka, and all experts attended physically. The validation protocol was designed based on the key guiding question of ‘How do epidemics and pandemics impact the multi-hazard preparedness and response in Sri Lanka?’

Experts were primarily selected based on their expertise, especially in the fields of public health, epidemiology, virology, infectious disease prevention and control, disaster risk reduction, sociology, curative health, and social research, which are relevant to the main aspects of epidemic, pandemic, and multi-hazard preparedness and response. The Ministry of Health, Sri Lanka, assisted in selecting suitable experts who are actively involved in COVID-19 and other epidemic preparedness and response activities in national operations.

In terms of selection criteria, each expert had substantial experience and expertise in areas directly related to public health, disaster management, or epidemiology, ensuring that their knowledge aligns with the multi-hazard context of epidemic and pandemic preparedness and response. The panel comprises a wide range of specialisations, from public health administration and infectious disease control to disaster risk reduction and social research, to provide comprehensive insights into developing a fully integrated system. Each expert held a key position within relevant institutions, such as the Ministry of Health, universities, or disaster management organisations, to ensure that their contributions were informed by current practical challenges and opportunities within the public health and disaster management sectors. Preference was given to experts who have demonstrated experience in collaborating across disciplines, especially in public health emergencies, to ensure that the framework benefits from integrated perspectives. The experts, their positions, and affiliations are further detailed in Table 1.

The experts were involved throughout the validation process. In the consultative meetings, expert panel members were briefed about the aim and objectives of the framework and the process of the development of the framework, definitions, working definitions of the components captured by the desk study and systematic review, and the framework validation method. All experts were divided into three groups and assigned three distinctive aspects of the framework mentioned below to discuss. The focus group discussions were used as the data collection technique. One member from each group recorded the outcome of the discussion and was briefed at the consultative meeting. Researchers recorded the briefing notes to be used as feedback for framework revisions. The groups addressed the following three aspects, and the final output of each group was recorded to improve/change the framework.

Are definitions and working definitions given to each sub-domain, component, and phase needing further revision?Do elements categorized under the distinct sub-domains relate and are they acceptable regarding recognized interventions?Does the drafted framework present logical flow and proper interlinkages of the components?

Ng’etich et al. [39] used this method to validate a framework developed to improve neglected tropical disease surveillance and response at sub-national levels in Kenya. Alharmoodi and Lakulu [38] and Calciolari et al. [40] have also used the same approach, using a survey questionnaire to collect data from experts during validation.

Section 4 elaborates on the framework validation process, including subsequent changes made to the framework during expert interviews.

## 3. Results

The findings of the study are structured according to the three guiding research questions: key elements of epidemic and pandemic preparedness and response, with a focus on COVID-19 derived from the literature review; key phases of the epidemic and pandemic that have been used in developing the conceptual framework; and the significance of the multi-hazard perspective within epidemic and pandemic preparedness.

### 3.1. Key Elements and Components of Epidemic and Pandemic Preparedness and Response

This section presents the key elements and components derived from the desk study on the epidemic and pandemic preparedness and response systems. Several sub-elements with different functions but close relationships with similar elements have been presented in the same component in the framework. For example, surveillance of pathogens, screening of raw data, and detecting pathogens are three different elements, but in the framework, these three have been amalgamated as a result of their close interconnection.

#### 3.1.1. Epidemiological Surveillance, Screening, and Detection

Early detection, preparedness, and response are primarily integrated into the epidemic and pandemic response processes, which are very similar to other disaster response activities [41]. However, an effective warning and dissemination process is the key to successful and efficient disaster response activities, including pandemic response [42]. The unpredictability of the emergence of new viruses and the re-emergence of pathogens makes preparedness and response very critical during an infectious disease outbreak. Most often, epidemics and pandemics arise from various origins, pathogens, and drivers, making their prevention and preparedness complex [43]. High-impact infectious disease outbreaks have been caused by viruses including SARS-CoV-02, Human Immuno Deficiency Virus (HIV), Ebola, Zika, and pandemic influenza. Similarly, most emerging infectious disease outbreaks are caused by bacteria (e.g., plague, anthrax, cholera, Q fever, typhus, rickettsial diseases), fungi, or helminths [44] (EU, 2020). However, early detection through diagnostics is key to outbreak containment and initiating outbreak management protocols [45,46,47].

Several studies highlighted that constant surveillance, detection, and timely dissemination of data, including genomic data of pathogens, are the most critical components of the early warning process that accelerates the response actions in any human or animal disease outbreak [10,48,49]. Many sudden larger outbreaks of the SARS-CoV-2 virus reported during the pandemic sparked the need for constant vigilance on the emergence and amplification of the virus to respond quickly and effectively. For example, months after becoming the first country in the world to test its entire population, the highest COVID-19 infection rate was reported in Slovakia, with a high death toll highlighting the significance of epidemiological screening [50]. Colson et al. [51] emphasized the importance of genomic surveillance and detection during a disease outbreak for immediate response and vigilance and noted the case of initial tracking of the IHU variant of SARS-CoV-02 reported from south-eastern France.

Climate change is a prominent risk driver that should be factored into public health systems regarding building preparedness and resilience to growing climate-related threats [52,53]. Studies have shown how climate change might influence the fragile balance between the human-animal interface, with the potential to drive the emergence of new pathogens [54]. The importance of integrating climate surveillance into public health surveillance systems to understand common drivers of infectious diseases has been largely highlighted [19]. Integrated surveillance systems enable the effective detection of combined drivers of pathogens. For example, in Sri Lanka, dengue rises yearly with monsoon showers due to an increase in mosquito breeding [55]. Further, vector-borne and helminths are identified as climate-influenced infections that create complications in tracking the combined origins of pathogens and sources [10,44]. Similarly, the incorporation of zoonotic disease surveillance, including transboundary animal diseases and aquatic-origin infections within the human-animal disease interface, strengthens the early detection and timely dissemination, which is required for improving robust outbreak responsiveness [49,56]. Therefore, strengthening human-animal-climate surveillance to track infections effectively when a pathogen emerges and sustaining early warning and dissemination among the countries will be of the utmost importance in controlling outbreaks during epidemics or pandemics.

#### 3.1.2. Risk and Vulnerability Assessments

Risk factors associated with a pandemic are understood as combined effects of spark risk and spread risk. Spark risk is the immediate risk of a pandemic, such as the emerging pathogen and the starting transmission. In contrast, the spread risk is driven by the larger cascading impacts of a massive pandemic outbreak [57]. Very common to other disasters, risk assessment enables understanding the risks and anticipated vulnerabilities. It guides the possible mitigation and preventive strategies, providing necessary information and data in a public health emergency. Risk assessment includes hazards, exposure, and vulnerability assessment methods [58]. The primary risk and vulnerability assessment is a critical component in an infectious disease outbreak since risks of potential outbreaks, the likelihood of intermediate hosts, and the possibility of human transmission are determined by the predictions. The core of the prediction is collecting and analyzing relevant information and raw data to explore spatiotemporal transmissions and epidemic laws of infectious disease, and models developed using genomics, statistics, and mathematical methods [48]. Like the risks, assessing vulnerabilities in an epidemic or pandemic is also essential to deciding on the appropriate response strategies. However, Jeleff et al [52] argue that understanding vulnerability is a very dynamic process due to the impact of a more significant number of social, cultural, and psychological factors in the local context. Therefore, tools developed for assessing vulnerability based on predetermined factors are believed to be less effective.

#### 3.1.3. Prediction, Alerts, and Early Warning

Effective prediction process and early warning processes provide significant guidance and direction to pandemic preparedness. These have been identified as core pillars of risk reduction during previous pandemic incidents [59,60,61]. However, the early warning process seems to be critical and dynamic and requires close coordination between relevant upstream (policy initiative) and downstream (implementation level of response and recovery) stakeholders for an effective pandemic response [62]. Risk governance in a pandemic context relies on a proper warning and dissemination process, a potential gap highlighted by several research studies during the COVID-19 pandemic [34,53].

Predictions, issuing alerts, and disseminating warnings to the public are critical elements in epidemic and pandemic early warning and response systems [10,48]. Prediction is a continuous and rigorous process that involves various statistical and data science models. The prediction process in a pandemic or epidemic comprises key prediction stages, including understanding the infectivity, transmissibility, agent, host, and environment dynamics, as well as morbidity and mortality trends, including case fatality rate [48].

Issuing alerts and early warnings is a critical measure in any infectious disease outbreak, which facilitates deciding the most appropriate control and preventive strategies before amplifying public transmission. Therefore, early warning requires strong coordination and networking from local levels to global partnerships in a timely manner. For instance, sharing genetic sequences and information via global networks enabled many countries to augment or accelerate their preparedness and response systems during the COVID-19 outbreak [56]. Emphasizing past human-animal infectious disease outbreaks, Tekola et al [49] highlighted the significance of screening raw data and information, triage of relevant information and risk assessments, alert/early warnings, and effective responses in understanding the essential elements of a functional early warning system.

Risk communication enables people to share information on emerging pandemics or epidemics, which guides people to take protective and preventive actions. The dissemination of basic and essential information associated with a pandemic, such as how the pathogen is transmitted, patient care, high-risk practices, and especially protective and preventive measures, enhances the effectiveness of disease control and mitigation [57]. However, risk communication and warning dissemination must be an exact process designed on the basis of evidence-based facts, which is fit for purpose and counters rumors and misinformation. Public trust in the message and the relevant authority is deemed to be highly critical when disseminating warnings in a public health emergency [44]. Many rumors are constructed based on sociodemographic factors like age, gender, personal ideologies, belief systems, knowledge, and cultural practices. As a global trend, people rely more on online sources and their trusted social networks [63]. However, a strong relationship between institutional trust, effective communication, and the mortality rate has been identified during the COVID-19 pandemic [64].

#### 3.1.4. Preparedness

Pandemic or epidemic preparedness is the key component of overall interventions toward disease outbreak-associated risk mitigation in any setting. Readiness to face a larger number of challenges caused by disease outbreaks, from onset detection to elimination, is included in a comprehensive pandemic or epidemic preparedness plan. In general, epidemic or pandemic preparedness is an emergency management planning framework containing key interventions aiming at reducing disease infection, such as epidemiological screening and projection and selection of public health prevention strategies and control measures [7]. Health emergency preparedness is an integrated and policy-based action plan implemented with intersectoral coordination from the national to the local level.

Public health emergency preparedness and response systems are comprised of legislation, which refers to a series of laws and regulations associated with public health emergencies, the organizational structure that is established to implement strategies and policies, and a response mechanism, which is the organization of functions towards the public health responsiveness [25]. Therefore, preparedness largely relies on a series of interventions addressing primary, secondary, and long-run implications underlying the cascading impacts of a pandemic outbreak.

McCabe et al. [65] have identified three elements (ready, willing, and able) in a framework that needs consideration in improving preparedness for public health emergencies. This RWA framework presents an interconnection of planning, implementing, and evaluating efforts to ensure high-quality individual and organizational responses to public health emergencies. Accordingly, readiness indicates that an individual or collective of individuals, agencies, and so forth is available for prompt action, service, or duty, and that an individual or collective possesses the human and material resources necessary for timely responses. The second element, willingness, refers to the state of being inclined or favorably predisposed in mind, individually or collectively, toward specific responses. The third, ability, refers to an individual, organization, or community’s actual operational power (i.e., skill, know-how) to perform a task if the requisite external circumstances require and allow it.

#### 3.1.5. Control and Mitigation

Control can be defined as reducing disease incidence, prevalence, morbidity, or mortality to a locally acceptable level due to deliberate efforts [66]. The primary aim of control and mitigation measures is to slow down disease transmission and minimize morbidity and mortality. Control and mitigation largely rely on wider decision-making in mitigating pandemic risks based on the information provided by risk and exposure assessments. Further, mitigation measures rely on various dimensions such as sociocultural, political, economic, epidemiology, and other necessary areas of interest that may be heavily impacted by pandemic outbreaks [67]. Many countries used different country-based mitigation measures during the COVID-19 pandemic while the vaccination process was implemented [68]. Physical distancing, quarantine, isolation of symptomatic cases, and contact tracing and testing were used by many countries as mitigation strategies. Kucharski et al. [69] investigated what combination of measures, including novel digital tracing approaches and less intensive physical distancing, might be required to reduce transmission. Teslya et al. [70] highlighted that information dissemination about COVID-19, which causes individual adoption of handwashing, mask-wearing, and social distancing, can be an effective strategy to control and mitigate disease outbreaks, and that short-term, government-imposed social distancing can buy time for healthcare systems to prepare for an increasing COVID-19 burden. Bruinen de Bruin et al. [67] identified clusters of risk mitigation measures implemented by various countries during the COVID-19 pandemic, ranging from mobility restrictions, socio-economic restrictions, physical distancing, hygiene measures, and communication to international support mechanisms. Wu et al [4] suggest that overall health interventions can be categorized under three broader categories, known as population-based interventions, such as lockdown, face masks, personal hygiene, and social distancing, and case-based interventions, like case detection, contact tracing, isolation and surveillance of confirmed cases, and border control measures, such as travel restrictions and mandatory quarantine requirement when travelling from a high-risk area. However, previous pandemics suggest that the size of the outbreak and pandemic impacts have been exacerbated mainly due to delays in applying proper control measures [57].

#### 3.1.6. Response

The rapid onset response primarily determines control of a pandemic outbreak once the initial detection is conducted. Many previous epidemics and pandemic events have highlighted the significance of early actions toward controlling the amplification of transmission immediately after confirming the onset record [49]. The response is an integrated system that combines all international, national, provincial, and community-led governance activities to mitigate pandemic-associated risks. Therefore, response management is part of the overall pandemic or epidemic control [7]. There are no common approaches or strategies to epidemic or pandemic response, which is determined by a country’s socio-demographic, economic, political, and epidemiological factors. Shaw et al. [71] note that the pandemic is global but the response is local, as almost all response measures, including medical countermeasures, are not necessarily universal. The response management system is believed to be a blend of the country’s emergency response approach, regulations, governance body, science-based decision making, and citizen behavior. By studying the response strategies of eight countries during the first wave of the COVID-19 pandemic, one study suggests that countries have used three key response strategies. These are aggressive containment, which aims to eliminate community transmission within 28 consecutive days; suppression, which aims to suppress and minimize community transmission by implementing public health interventions; and mitigation, which seeks to avoid overwhelming health systems by flattening the epidemic curve or achieving herd immunity in the population, which is a long-term approach using a set of extensive interventions [4].

#### 3.1.7. Primary Phases of Epidemics and Pandemics: From Emergence to Endemic

The WHO [10] has identified a framework containing four key phases of an epidemic in terms of understanding the different stages of spreading infectious disease and the magnitude of the outbreak. The subsequent interventions that should be executed as possible response strategies to the respective outbreak stages have also been analyzed in the aforementioned framework. Accordingly, the first phase is the introduction or emergence of the pathogens, where anticipation and early detection are critical in epidemic response. The emergence of a pathogen is believed to be unpredictable. However, anticipation enables health systems to focus on diseases most likely to emerge and drivers to quickly identify when a pathogen has emerged. Early detection facilitates taking necessary early actions to avoid spreading the disease to a more significant extent. Then, the localized transmission phase occurs, where human transmission gradually increases in local territories but cannot be seen in an intense nature. In the third phase, the outbreak amplifies with a high transmission rate, causing an epidemic or pandemic—the pathogen can transmit from human to human and causes a sustained outbreak in the community, threatening to spread beyond the local territories. The fourth phase is reduced transmission, which is the stage where human-to-human transmission of the pathogen decreases mainly due to acquired population immunity or effective interventions to control the disease outbreak [10].

However, the four phases discussed in the WHO [10] are limited to epidemics. According to Antia and Halloran [72] and Lavine et al. [73], the ‘endemic phase’ has been widely discussed by scientists as a transitional phase of global pandemics, assuming that endemic would be the next stage of the COVID-19 pandemic. As highlighted in the conceptual framework (Section 4.1), a pandemic to endemic transition of infectious disease is a real possibility in the long run. Given the infectivity, virulence, emergence of the variants, transmission dynamics of the disease, and availability of effective vaccines, eradication seems highly unlikely. Realistically, the goal will be to achieve endemicity, which is the stable maintenance of the pathogen with a lower disease prevalence. Such endemicity has been observed before among other coronavirus types [73].

However, the path to endemicity is dependent on several factors. Once an epidemic or pandemic is established, the ‘Basic Reproduction Number’ or R_0_, which denotes the number of secondary infections produced by an infected individual when the population is entirely susceptible to the disease, is usually greater than one [72]. Subsequently, the exponential growth of cases can be expected since all three components of the epidemiological triad, the agent, host, and environmental factors, facilitate such growth.

Nevertheless, this exponential growth would not last for an extended period due to the fact that there are fewer susceptible individuals in the population. This phenomenon is caused by the majority of the population being infected with the primary infection and acquiring temporary immunity or immunity achieved through vaccination. Hence, as the epidemic progresses, the ‘Effective Reproduction Number’ or R_eff_ falls and the epidemic subsides [72]. Unless the virus goes extinct, which is unlikely in the case of respiratory viruses like SARS-CoV-2, the infection enters a phase of endemicity with a low prevalence of the disease. The R_eff_ remains equal to one on average; however, the appearance of disease outbreaks from time to time remains a possibility. The birth of individuals not exposed to the virus, immigration, and the waning of previously acquired immunity could ignite another outbreak when the environment is conducive [73].

Using the work of the WHO [10], Antia and Halloran [72] (2021) and Lavine et al. [73] (2021), the following five phases were finally identified as the basis for any epidemic or pandemic.

Introduction or emergenceLocalized transmissionAmplificationReduce transmissionEndemic

### 3.2. Key Interventional Stages

The following three key stages were identified for understanding the scope of medical and non-medical interventions in epidemics and pandemics.

UpstreamInterfaceDownstream

The upstream-downstream metaphor, initially recognized by medical sociologist Irvin Kenneth Zola in analyzing micro- and macro-level determinants of health outcomes, is a widely recognized parable in health promotion interventions [74]. This metaphor has been largely used by scholars to discuss ‘structures’ and ‘determinants’ of social inequalities of health in understanding ‘symptoms’ (downstream) and ‘causes’ (upstream) of health inequality as a complex dichotomy [75]. According to Dopp and Lantz [76], macro-level interventions, such as government and policy structures, operate upstream, and individual-level micro-interventions operate downstream. Gehlert et al. [77] have highlighted the complex interconnection between downstream and upstream, analyzing determinants of multiple health disparities among breast cancer patients in their study. However, recognizing an intermediate interventional level that operates between upstream and downstream in this conceptual metric, ‘midstream’ has been separately identified by health scholars focusing more on mezzo-level interventions [76]. Accordingly, the interconnection between the upstream, midstream, and downstream in terms of interventional point of view can be summarized, and is shown below in Figure 2.

Sakalasuriya et al. [78] have used the upstream-downstream metaphor to analyze key components of a conceptual framework focused on end-to-end tsunami early warning and mitigation systems from a DRR perspective. According to the study, monitoring and detection occur upstream, whereas information dissemination occurs downstream [78]. For DRR, upstream includes detection, verification, threat evaluation, and forecasting. The downstream mechanism includes evacuation, delivery of public safety messages, initiation of national measures, and preparation and implementation of standard operations [62,78]. Moreover, Sakalasuriya et al. [78] have used the term ‘interface’ to understand the linking stage between upstream and downstream, which is dedicated to decision-making on evacuation and issuing alerts and early warnings in understanding the response process particular to tsunami evacuation.

According to the two concepts, midstream refers to various risk drivers and risk behaviors that need controlling by executing preventive strategies amid any public health emergency. In contrast, the interface refers to a set of involvements related to early warning, which is an integral part of risk communication in an emergency. Therefore, the term interface is used in the conceptual framework of this study to identify interventions for decision-making and early warnings for epidemics and pandemics.

### 3.3. Multi-Hazard Preparedness and Response within Epidemic and Pandemic Context

The need to incorporate a multi-hazard perspective in public health emergencies has been sparked by the dozens of natural and manmade health hazards reported in several parts of the world during the COVID-19 pandemic [79]. Multi-hazards can be defined in two ways: (a) The selection of multiple significant hazards that the country faces, and (b) the specific contexts where hazardous events may co-occur, cascadingly or cumulatively, over time, and taking into account the potential interrelated effects [80].

During the multiple hazards experienced by many countries amidst the COVID-19 pandemic, risks and impacts were largely cascading (Quigley et al., 2020a). The Pacific and North Pacific Islands, especially Vanuatu and Palau, were hit by the Harald cyclone and a long-run drought amidst the pandemic. In contrast, Indonesia was severely impacted by heavy rain and floods during the peak of the COVID-19 outbreak [14]. The Congo experienced multiple challenges due to the concurrent Ebola outbreak and the COVID-19 pandemic [17]. Roth [81] highlighted that the same scenario was reported in Guinea due to Ebola and COVID-19 occurring simultaneously. India also faced multiple challenges due to the Amphan cyclone and mucormycosis, known as the black fungus, which occurred during the COVID-19 outbreak [82]. Further, in 2021, European countries were affected by flooding, while Malaysia also evacuated 30,000 people due to monsoon flooding amid the COVID-19 pandemic [83]. The La Palma volcanic eruption on Spanish Island and earthquakes caused more than 6000 people to evacuate and ruined agriculture-based livelihoods in the area [84].

Sri Lanka, where the conceptual framework was validated in this research, is also a multi-hazard-prone country that has faced various challenges caused by concurrent hazards amidst the COVID-19 pandemic. For example, in 2020, Sri Lanka experienced a prolonged drought in the dry zone, and with the monsoon rainfall, the country faced severe flooding in many districts. Concurrently, cyclone ‘Buravi’ also hit the country while the COVID-19 pandemic was at its peak in the same year [16]. In 2020, several malaria-infected people were also found in Sri Lanka [85]. Again, the heavy monsoon hit ten districts in 2020, especially the southern part of the country, causing a significant flood event, which led to 245,212 people being displaced and 16 deaths within a month [86]. Meanwhile, with the rise of monsoon rainfall, dengue cases became a critical juncture. This seems to be a yearly pattern after the monsoon showers [16].

Having understood the multiple risks and challenges posed by concurrent hazards, countries need to transition to national emergency preparedness plans or DRR frameworks that are more focused on multi-hazard preparedness. Linares et al. [87] stress the need for integrative preparedness to tackle multiple risks and impacts on public health, which is believed to be increased due to climate change. This necessity is highlighted by the UNDRR [79], which emphasizes the significance of understanding climate emergencies and the systematic impacts of the COVID-19 pandemic in formulating policy measures for reducing risks of uncertainties.

It is crucial to integrate several components of epidemic and pandemic preparedness with multi-hazard preparedness systems. Surveillance and detection of pathogens, for instance, are key components that have a reference to other drivers related to natural hazards, such as climatic factors (i.e., drought, rain). This interconnectedness underscores the importance of integrating climatic surveillance with the public health surveillance system [1,88]. This possibility has been tested during the COVID-19 outbreak [89]. Similarly, issuing early warnings and alerts during a public health emergency is very common, and it can combine with other early warning systems, forming integrated early warning systems [87]. Further, risk assessment has immense potential as multi-hazard risk assessments, of which the multi-hazard risk index is a well-established strategy used by many hazard-prone countries [90,91]. Even though prediction and decision-making, which guide the response and control, are hazard-specific, a multi-hazard approach is necessary due to the potential of occurring concurrent hazards. Therefore, the multi-hazard perspective has been recognized as a critical component that needs to be incorporated into preparedness and response systems dedicated to epidemics and pandemics in any context.

## 4. Discussion

### 4.1. Conceptual Framework for Epidemic and Pandemic Preparedness and Response

Understanding policymaking during a global health emergency and prioritizing the safeguarding of citizens is a critical aspect for any country, as seen during the COVID-19 pandemic [92,93]. Analyzing the potential of practices, strategies, and interventions for disease prevention and control thus supports policymaking in preparedness and response [67,68]. Several frameworks suggested that comprehension of dynamics on preparedness and response for a global health emergency, such as COVID-19, is pivotal due to the complexity of their impact on the larger social and health systems [13,20,30,67]. Many such frameworks lacked overall elements and components of a preparedness and response system dedicated to epidemics and pandemics. Recognizing this gap, this study proposes a comprehensive framework that elaborates on key elements of an epidemic and pandemic preparedness system, which can be used to formulate policies on disease prevention and control during a large disease outbreak. In addition, we have incorporated the critical aspect of concurrent hazards that was largely missing in the research literature.

Overall, this section discusses the key elements derived from the systematic review analyzed in Section 3.1, Section 3.2, and Section 3.3. Also, the elements identified to develop the framework have been categorized into three main principal components according to their functions and scope within the epidemic and pandemic preparedness ecosystem. Accordingly, three possible intervention stages, five pandemic phases, and ten key elements of epidemic and pandemic response interventions were identified. Definitions and working definitions for all elements have been provided in Table 2. Standard definitions were obtained from the WHO, Centers for Disease Control and Prevention (CDC), and other sources identified through the desk study and systematic review. Further, several working definitions have been formulated and modified based on the facts revealed by the desk study analysis and expert views obtained in the validation process.

Accordingly, the following detailed framework (Figure 3) was developed based on the analysis of key elements and stages identified through literature. Rationale and interconnectedness between components were analyzed through the literature survey to arrange key elements and stages in the framework.

All key components were categorized into three broader sub-domains, known as upstream, interface, and downstream, according to their relevance and interconnections to reflect a wider perspective of preparedness and response systems. The following subsections describe the main components of the framework for clarity.

Upstream-downstream-interface: The upstream mechanism is more involved in activities and policies related to the epidemic and pandemic preparedness to mitigate and reduce risk and vulnerabilities. Accordingly, surveillance and detection, risk assessment, and preparedness are the key elements of the upstream mechanism. Downstream is highly reliant on response and recovery, especially referring to the interventions that affect disease control and mitigation at the grassroots level. The interface is the dedicated interventional phase for alerts and early warning, followed by prediction and decision-making on the disease outbreak.Key connections of elements: according to the features presented in the framework, epidemiological surveillance and detection are the first components that should be primarily interlinked with both zoonotic surveillance and climate surveillance for detecting zoonotic origin pathogens and infections like vector-borne and helminths, which are likely to emerge with changes in climatical conditions. However, surveillance and detection are often aimed at diagnosing seasonally or newly emerging pathogens. This process involves a cautious scientific data screening process to identify new pathogens, including their source of origin. Nevertheless, surveillance relies more on careful monitoring and anticipation of pathogens likely to emerge, and therefore constant vigilance is a must to forecast the patterns of emerging infections and their drivers.Preparedness for a pandemic or epidemic is known as the readiness point, which executes imposing preventive strategies combined with medical countermeasures such as vaccine/prophylaxis and non-medical countermeasures including social and physical distancing, self-isolation, travel bans (mobility restrictions) and restriction, disinfection measures, and quarantine. However, the response stage, which is the next element, is largely established by a range of control and mitigation strategies to be identified based on the magnitude of the disease outbreak. Hence, varieties of response strategies are deemed to be altered mainly on the level of morbidity and mortality, infectivity, transmissibility, and the emergence of variants of concern that are encountered in different stages in a pandemic, measured by a variety of indicators, including case fatality rate (CFR).Preventive strategies and multi-hazard preparedness and response: two subsequent processes must be cohesively and systematically integrated from the beginning to the endpoint of the preparedness and response process. Further, as the framework reflects, both preventive strategies and multi-hazard preparedness consist of a range of subsequent interventions throughout the preparedness and response eco-system. Preventive strategies are largely determined by the respective stages and the size of the disease outbreak once it emerges. For instance, preventive interventions adopted in the localized transmission stage could be considerably distinctive from the strategies devised at the amplification stage, where disease outbreak is quickly ignited.Transition to Endemic: having understood the key phases revealed by the WHO in explaining infectious disease outbreak scenarios, the bottom tier, which reflects the disease outbreak phases, has been extended to the pandemic to endemic transition stage, which is more likely to happen at the end of a more significant pandemic outbreak due to factors such as sustained public health countermeasures and vaccination. An epidemic is an increase, often sudden, in the number of cases above what is usually expected in a population of a particular geographical area, whereas a pandemic denotes a transborder or worldwide spread of a particular disease. The transition from pandemic to endemic is associated with several aspects, including high immunity levels/achieving herd immunity and factors related to the agent (virus) itself. These factors are widely recognized as agent factors (reduced virulence and a smaller number of mutations and new variants), host factors (improved immunity, herd immunity, reduced susceptibility), environmental factors such as modifications in the living environment due to mobility restriction, and strict public health security measures. As per the epidemiological data, this can also be possible with the COVID-19 pandemic [72].Therefore, this transition from epidemic to endemic involves careful consideration of many factors, including susceptible age cohorts, the status of immunity and herd immunity, agent/host/environment equilibrium, sustaining public health security measures, and effective rollout of vaccination. As highlighted in the conceptual model, it is important to sustain and continue preventive strategies throughout the entire process so that any chance of the reemergence of an epidemic is prevented.Transition to new normal: lessons learnt from the epidemic or pandemic must be systematically integrated into the future capacity building for health system resilience. In the post-pandemic stage, learning to build resilience for similar occurrences is very important. Therefore, incorporating lessons learned and best practices into the existing public health measures, particularly policy formulations, must be a cyclical process in bridging the experience of any epidemic or pandemic into the prevailing preparedness and response frameworks. Every disease outbreak has many lessons to consider regarding policy changes or formulating new policy measures [4]. Transforming policies and practices to better prepare or ‘build back better’ for the next event is worthwhile. Therefore, the connection of the lessons learnt from an epidemic or a pandemic to policy transformation is also highlighted in the framework.

### 4.2. Validation of the Conceptual Framework

#### 4.2.1. Phase 01

The conceptual framework presented in Section 4 identified key phases of the epidemic and pandemic and components of the preparedness and response system in general. Further, the framework describes the connection of elements to the respective epidemic and pandemic phases, focusing on the COVID-19 pandemic. Thus, this framework primarily applies to understanding the phases and components of any global epidemic or pandemic preparedness and response system. However, the conceptual framework has been validated on the epidemic and pandemic preparedness and response system in Sri Lanka, focusing on a multi-hazard context. During the validation, natural and health hazards, i.e., floods, cyclones, dengue, and leptospirosis, concurrently encountered amidst the COVID-19 in Sri Lanka, were considered to understand the complexity of multi-hazard preparedness.

The initial draft of the framework was a linear model, and components associated with epidemic and pandemic preparedness and response have been arranged from surveillance and detection of pathogens to eradication or elimination of the disease [19]. This framework was presented on 20 September 2022 at the first expert discussion held in Colombo. Researchers briefed the summary of papers used for capturing elements for the framework at this forum, as experts could not read the selected papers. The primary suggestion from the discussion was the possibility of understanding the major components of response interventions and arranging the elements within broader generic components rather than presenting a leaner process of epidemic and pandemic preparedness and response.

The second version of the framework was then revised according to the results of the first expert discussion and improved based on an upstream-downstream-interface model, which has been identified by several research teams [62,78] as referring to a conceptual framework to analyze intervention phases of tsunami early warnings. Key elements identified by the systematic review were then placed in upstream, interface, and downstream mechanisms based on the key functions associated with preparedness, early warnings, and response, informed by studies published on an upstream-downstream-interface model [62,78]. This revised version was presented at the second expert discussion. Key elements assembled in the upstream-downstream interface were rearranged according to the views and suggestions of the expert panel. Several sub-elements that are associated with the same preparedness and response interventions were merged and included in the framework to maintain the clarity of the content. For example, epidemiological surveillance, detection of pathogens, and primary screening were initially identified separately. However, they were combined as these three are primarily linked with disease surveillance.

In the third discussion, preventive strategies, which were embedded downstream, were identified as a significant element that requires consecutive execution throughout the entire preparedness and response process as a key measure, the same as multi-hazard preparedness. Further, the term eradication, which was placed downstream, was replaced with the term ‘interventions toward elimination’, as suggested by experts in epidemiology and public health, based on the fact that many infectious diseases cannot be completely eradicated due to re-introduction or re-emergence.

In the final validation workshop held in Colombo, Sri Lanka, on 12 January 2023, the pandemic to endemic transition was added to the epidemic phases, which have been initially identified by the WHO [10]. The endemic phase was informed by the work of Antia and Halloran [72] and Lavine et al. [73] and suggestions of public health experts, particularly in relation to the COVID-19 pandemic. Moreover, the significance of incorporating lessons learned from any epidemic or pandemic into the policy measures that highly rely on decision-making in upstream was also embodied in the framework in this final validation point.

#### 4.2.2. Phase 02

The preliminary framework was presented at two international research conferences on public health and disaster resilience, which were sparked the need for research and innovations to address the global challenges generated by the COVID-19 pandemic. Presenting draft frameworks at scientific sessions and conferences to obtain inputs from various experts and stakeholders has been recognized as a potential and rigorous validation strategy by research conducted on framework validation studies [39,40].

The initial framework was presented at the second international symposium on disaster resilience and sustainable development, held at the Asian Institute of Technology, Thailand, on 24–25 June 2021 [105,106] to obtain views and suggestions from international experts who participated in the discussion. This session was dedicated to Public Health and COVID-19 Risks. The recommendations made by the researchers and policy experts on the terminology, definitions, and contradictions seen among selected elements in the draft framework were tabled at the validation meetings to consider changes in the framework.

The refined framework was presented at the International Research and Innovation Symposium on Dengue Amidst the Pandemic: Improving Preparedness and Response for Multi-hazard Scenarios held on 16–17 March 2022 at Colombo, Sri Lanka [107]. This conference was a hybrid conference, which facilitated broad participation of academics and practitioners, especially in public health, epidemiology, virology, infectious disease prevention, malariology, health care management, and various other health and DRR-related fields.

#### 4.2.3. Limitations and Applicability of the Study

In terms of the systematic review, literature sources were selected that focused on the richness of elements and components of the COVID-19 pandemic during literature mining. Therefore, characteristics of other disease outbreaks within the same period, i.e., Ebola, may have been reflected inadequately. Another limitation of the study is that the framework has only been validated in the Sri Lankan context and hence needs validation in other settings. Also, the framework captures only the major components of the pandemic preparedness and response system. In addressing this gap, future studies can be aimed at developing a detailed framework that captures all sub-elements in the preparedness and response process, which may be specific to disease events. Also, some disease outbreaks may not complete all the phases mentioned in the framework due to several epidemiological factors, including building herd immunity among the population, effective mitigation, and control, such as producing effective vaccines, etc.

This framework has already been implemented during dengue outbreaks in Sri Lanka to ensure its wider applicability. Early disease identification through surveillance, case notification, timely patient referral for early treatment, and community awareness for prevention align with the approaches identified in the conceptual framework.

## 5. Conclusions

As the COVID-19 pandemic revealed fresh insights and challenges, our orthodox DRR perspectives and methods should be re-evaluated to understand the complexity and magnitude of forthcoming implications in the public health sector. The above framework on epidemic and pandemic preparedness and response highlights the need to expand existing understanding of the factors, determinants, and key elements associated with governance, preparedness, and response in terms of public health emergencies. However, the elaboration of the elements in the framework is only limited to the major interventions and components of the preparedness and response system, which helps to understand the systematic progression and interconnectedness of fundamental elements involved in the process.

Unlike the many epidemic and pandemic-related frameworks, this framework reflects multi-hazard preparedness, a key area that must be incorporated within the epidemic and pandemic preparedness and response process. The fact that none of the biological hazards would be an isolated event, as several other compounding hazards could have co-occurred amidst an epidemic or a pandemic, was evident during the COVID-19 outbreak. Therefore, the framework highlights that preparedness and response should not be limited to the single-disease event, which focuses only on risks and vulnerabilities associated with the disease outbreak. Most often, the single-event approach neglects cascading impacts that occurred due to the combined hazard events that would be more devastating than the epidemic or pandemic. Therefore, the framework suggests multiple processes that need consideration in mitigating the challenges of an epidemic or pandemic.

This framework would provide a structured and systematic approach to addressing diseases of pandemic and epidemic potential. Given all the unknowns at the onset of such a disease, such a methodical approach, prioritising the critical elements like surveillance, early detection, isolation/quarantine, treatment, etc., would avoid duplication of work by different stakeholders, saving much-valued resources in a resource-poor setting. However, the practical implications are not limited to resource-poor settings alone. Even in a well-developed health system, a methodical approach in addressing diseases of pandemic or epidemic potential would enable timely interventions, allowing the healthcare administrators to divert the right resources to the population cohorts that most require them.

## Figures and Tables

**Figure 1 ijerph-21-01238-f001:**
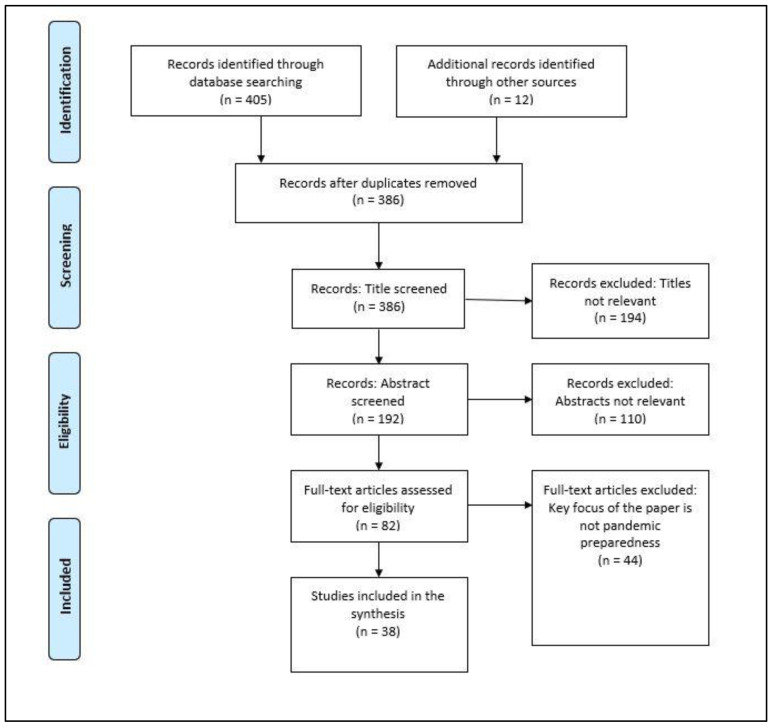
Literature mining process through electronic databases.

**Figure 2 ijerph-21-01238-f002:**
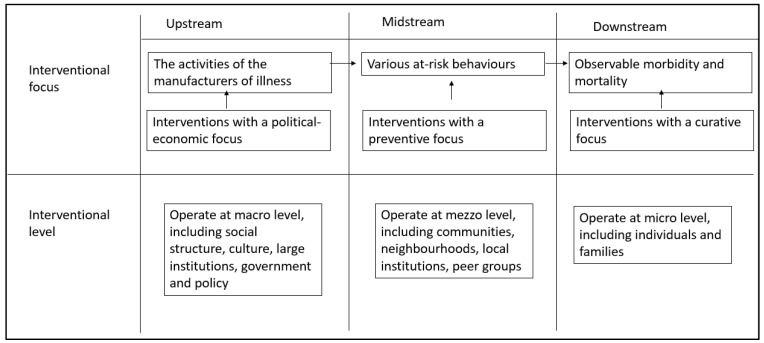
Upstream-midstream-downstream model used in health promotion. Source: [74,76], and authors’ composition.

**Figure 3 ijerph-21-01238-f003:**
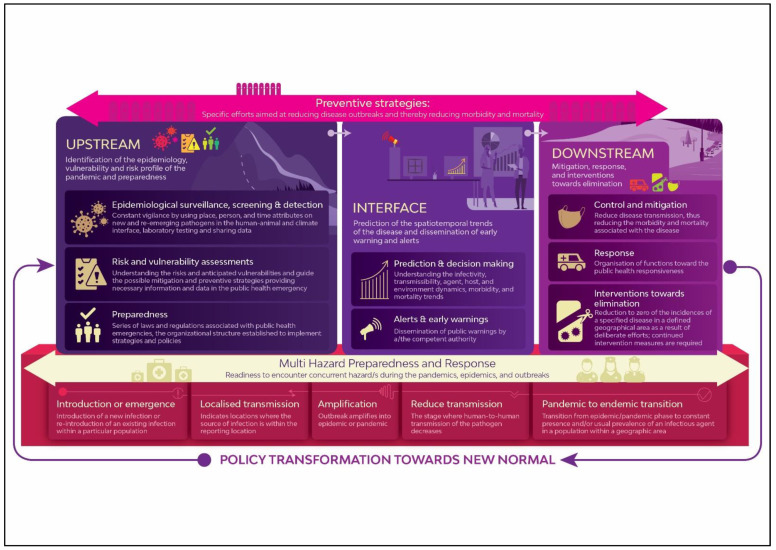
Framework for Epidemic and Pandemic Preparedness and Response. (Source: authors’ composition).

**Table 1 ijerph-21-01238-t001:** List of experts participated in the framework validation.

Position	Affiliation	Expertise
Additional secretary (medical services)	Ministry of Health, Sri Lanka	Public health administration, hospital administration and management
National coordinator, health sector disaster management	Ministry of Health, Sri Lanka	Public health, DRR
Acting director, Anti-malaria campaign	Ministry of Health, Sri Lanka	Public health, Infectious Diseases, Malariology
Director	National Dengue Control Unit, Ministry of Health, Sri Lanka	Epidemiology, Public health
Director (planning)	Ministry of Health, Sri Lanka	Public health administration, health systems strengthening
Senior consultant community physician	National Dengue Control Unit, Ministry of Health, Sri Lanka	Epidemiology, Infectious Disease Control, Risk Communication in infectious disease outbreaks
Public health registrar	National Institute of Health Sciences, Sri Lanka	Public health, Disaster management
Disaster management focal point medical officer	National Dengue Control Unit, Ministry of Health, Sri Lanka	Disaster Management, Public Health, Infectious disease control and prevention
Two Professors	Global Disaster Resilience Centre, University of Huddersfield	Disaster Risk Reduction
Professor	University of Colombo, Sri Lanka	Sociology, Disaster Management
President (elect)	Association of Disaster Risk Management Professionals of Sri Lanka (ADRiMP)	Disaster Risk Management, Civil Engineering
Senior Lecturer	University of Peradeniya, Sri Lanka	Sociology, Disaster Risk Reduction
Senior Medical Officer	Ministry of Health, Sri Lanka	Curative Health, Infectious Disease
Two Researchers	Social Policy Analysis Research Centre, University of Colombo	Social Research, Disaster Management
Researcher	University of Moratuwa	Civil Engineering, Disaster Management

**Table 2 ijerph-21-01238-t002:** Standard and working definitions for selected elements and phases used in the framework.

Element or Phase	Definition/Working Definition	Source
Interventional stages		
Upstream	Identification of the epidemiology, vulnerability, and risk profile of the pandemic and preparedness	The term and the concept were initially identified by Gehlert et al., [77]; Mckinlay [74] and modified in the validation process to be compatible with epidemics and pandemics.
Downstream	Mitigation, response, and interventions toward the elimination	
Interface	a set of involvements related to early warning, which is an integral part of the risk-communication in an emergency	The terms and the concept were initially identified by Haigh et al. [62]; Sakalasuriya et al. [78] and modified in the validation process to be compatible with epidemics and pandemics
Epidemic and pandemic phases		
Introduction or emergence	introduction of a new infection or re-introduction of an existing infection within a particular population	WHO [10]
Localised transmission	Indicates locations where the source of infection is within the reporting location	
Amplification	Outbreak amplifies into epidemics or pandemics.	
Reduce transmission	The stage where human-to-human transmission of the pathogen decreases	
Endemic	The transition from epidemic/pandemic phase to constant presence and/or usual prevalence of an infectious agent in a population within a geographic area	CDC [66]
Elements Associated with Preparedness and Response Interventions		
Epidemiological surveillance, screening, and detection	Constant vigilance by using place, person, and time attributes on new and re-emerging pathogens in the human-animal and climate interface, laboratory testing, and sharing data	Wolitski et al. [94]; Yeager et al [95]
Risk and vulnerability assessment	Understanding the risks and anticipated vulnerabilities and guiding the possible mitigation and preventive strategies by providing necessary information and data in the public health emergency	WHO [96]
Preparedness	Series of laws and regulations associated with public health emergencies, and the organizational structure established to implement strategies and policies	He et al. [25]; McCabe et al. [65]; Nelson et al. [97]
Prediction and decision making	Understanding the infectivity, transmissibility, agent, host, and environment dynamics, morbidity, and mortality trends	Anderson et al. [68]; Mamo [98]
Alerts and early warnings	Dissemination of public warnings by a/the competent authority	Xiong et al. [99]; Zhang et al. [100]
Control and mitigation	Reduce disease transmission, thus reducing the morbidity and mortality associated with the disease.	Burrell et al. [101]; McCloskey et al [102]
Response	Organization of functions toward public health responsiveness	Tan et al. [103]
Elimination	Reduction to zero of the incidences of a specified disease in a defined geographical area as a result of deliberate efforts; continued intervention measures are required	Walter [104]
Preventive strategies	Specific efforts aimed at reducing disease outbreaks and thereby decreasing morbidity and mortality	Burrell et al. [101]; Collett et al. [105]
Multi-hazard preparedness and response	Readiness to encounter concurrent hazard/s during pandemics, epidemics, and outbreaks	Quigley et al. [1]

## Data Availability

No new data were created or analyzed in this study. Data sharing is not applicable to this article.

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
