# Peer review of "Epidemic and Pandemic Preparedness and Response in a Multi-Hazard Context: COVID-19 Pandemic as a Point of Reference"

_ijerph, 2024, doi:10.3390/ijerph21091238_

Round 1
Reviewer 1 Report
Comments and Suggestions for Authors
Thank you for sharing your article on the epidemic and pandemic preparedness and response in a multi-hazard context. I have just the one very minor comment:
L119-122: Is the aim of this paper, stated within these lines, related to a specific pathogen such as COVID-19 or rather generic and pathogen-unspecific? Please clarify within your manuscript. Based on L159-160 it is related to COVID-19, no?
Comments on the Quality of English LanguageOverall the manuscript reads well. There are just minor issues where English editing would ease the readability of the manuscript.
Author Response
Comments 1: [L119-122: Is the aim of this paper, stated within these lines, related to a specific pathogen such as COVID-19 or rather generic and pathogen-unspecific? Please clarify within your manuscript. Based on L159-160 it is related to COVID-19, no?]
Response 1: [This has been clarified (see newly added lines 94-96)]
Comments 2: [Overall the manuscript reads well. There are just minor issues where English editing would ease the readability of the manuscript. }
Response 2: [addressed satisfactorily)
Reviewer 2 Report
Comments and Suggestions for Authors
The manuscript ijerph-3162992, by Kamalrathne and colleagues, addresses a critical and timely issue by developing a framework to enhance preparedness and response to pandemics in a multi-hazard context. The identification of key elements such as epidemiological surveillance, risk assessments, and preventive strategies highlights essential components necessary for comprehensive preparedness and response systems. The integration of these elements into three interventional phases (upstream, interface, and downstream) provides a structured and holistic approach that can be valuable for policymakers and public health officials. Despite these strengths, the study has several areas that require further attention and improvement. For instance, the framework presented in the study lacks detailed information on its practical application. Case studies or examples of successful application in different regions or contexts would significantly enhance the utility and impact of the proposed framework. In addition, the involvement of 17 experts in the validation process is commendable, but the study does not provide sufficient details on the selection criteria for these experts or the validation process itself. More transparency in this area would strengthen the credibility of the findings. Other points also require revision as follows:
- Lines 101-104/264-266/705-708/788-791: The sentences "Previous pandemic events have been identified and analysed to capture possible interventions and elements in the preparedness and response process for biological hazards have been identified and analysed", "The framework validation process and the subsequent changes that were made in the framework elaborated in section 6, to present the framework development from the beginning to the validation in a consistent process", "The transition from pandemic into endemic is associated with several, including high levels of immunity /achieving herd immunity as well as factors associated with the agent(virus) itself" and "The initial framework presented at the session on public health and COVID-19 risk of the second international symposium on disaster resilience and sustainable development, held at the Asian Institute of Technology, Thailand held on 24-25 June 2021 (Kamalrathne et al., 2021) to obtain views and suggestions of international experts participated in the discussion" require rephrasing for clarity.
- Line 498: The reference to Figure 2 should be placed at the end of the sentence on line 552.
- Lines 521-522: In the sentence "winning of previously acquired immunity could ignite another outbreak", the term "winning" should be replaced with "waning".
- Line 553: The expression "manufacturers of illness" in Figure 2 requires revision.
- Lines 646-647: According to the text on lines 570-571, the definition of "interface" in Table 2 should be "a set of involvements related to early warning, which is an integral part of the risk-communication in an emergency", instead of repeating the definition of "localised transmission" (i.e., "indicates locations where the source of infection is within the reporting location").
Comments on the Quality of English LanguageModerate editing of English language required.
Author Response
Comments 1: [Despite these strengths, the study has several areas that require further attention and improvement. For instance, the framework presented in the study lacks detailed information on its practical application.}
Response 1: [well addressed this suggestion by including its practical aspect from the Sri Lankan context ( see newly added lines 834-837)]
Comments 2: [In addition, the involvement of 17 experts in the validation process is commendable, but the study does not provide sufficient details on the selection criteria for these experts or the validation process itself. More transparency in this area would strengthen the credibility of the findings. ]
Response 2:[ A new section was added to clarify this selection (see lines 227-238)]
Comments 3: [- Lines 101-104/264-266/705-708/788-791: The sentences "Previous pandemic events have been identified and analysed to capture possible interventions and elements in the preparedness and response process for biological hazards have been identified and analysed", "The framework validation process and the subsequent changes that were made in the framework elaborated in section 6, to present the framework development from the beginning to the validation in a consistent process", "The transition from pandemic into endemic is associated with several, including high levels of immunity /achieving herd immunity as well as factors associated with the agent(virus) itself" and "The initial framework presented at the session on public health and COVID-19 risk of the second international symposium on disaster resilience and sustainable development, held at the Asian Institute of Technology, Thailand held on 24-25 June 2021 (Kamalrathne et al., 2021) to obtain views and suggestions of international experts participated in the discussion" require rephrasing for clarity.]
Response 3: [rephrased all suggested sentences]
Comments 4: [Line 498: The reference to Figure 2 should be placed at the end of the sentence on line 552.]
Response 4: [corrected ]
Comments 5: [ Lines 521-522: In the sentence "winning of previously acquired immunity could ignite another outbreak", the term "winning" should be replaced with "waning".]
Response 5: [corrected]
Comments 6: [- Line 553: The expression "manufacturers of illness" in Figure 2 requires revision]
Response 6: [The cited research has used this term, and thus, authors did not change the term as the original work has been cited]
Comments 7: [- Lines 646-647: According to the text on lines 570-571, the definition of "interface" in Table 2 should be "a set of involvements related to early warning, which is an integral part of the risk-communication in an emergency", instead of repeating the definition of "localised transmission" (i.e., "indicates locations where the source of infection is within the reporting location").]
Response 7: [Replaced the phrase]
Comments 8: [Moderate editing of English language required.]
Response 8: [Language moderately improved throughout the manuscript]
Reviewer 3 Report
Comments and Suggestions for Authors
Overall Comments
Thank you for the opportunity to review this paper. I appreciate the effort put into addressing a critical area of public health and emergency management. The combined use of systematic analysis and desk study methodologies is commendable, as it allows for a comprehensive review and the development of a conceptual framework. However, several areas could benefit from further refinement.
Introduction
Comment 1: The data and citations used in the introduction are outdated. Updating the introduction with more recent references would help set the context more clearly and reflect the latest research in this field.
Comment 2: The introduction dedicates too much space to general information about COVID-19. This section could be shortened, and more space allocated to reviewing literature directly related to the research question.
Comment 3: In lines 121-122, the paper mentions “a fully integrated system for epidemic and pandemic preparedness and response in a multi-hazard context.” This statement should be supported by reviewing additional research and previous frameworks to provide a stronger foundation.
Comment 4: The introduction could be more specific in outlining the exact gaps the study intends to fill. Clearly articulating the rationale behind the research questions would provide better context for the study.
Methods
Systematic Review: Comment 5: The search terms and strategies are not detailed enough. Including the date range, search terms, and any Boolean operators used would enhance the transparency and replicability of the review.
Comment 6: The exclusion criteria need more detail. Providing a clearer rationale for excluding certain studies would improve the methodology's robustness.
Comment 7: Lines 186-187 mention that six grey literature sources were included in the literature review. The criteria for selecting these sources should be clearly explained to justify their inclusion.
Comment 8: The paper should detail the exact variables extracted from each study and the process used to develop the framework or variables during the systematic review stage. This would enhance the study's transparency and replicability.
Desk Study: Comment 9: The methodology for the desk study should be more detailed. The criteria for selecting documents and reports should be clearly outlined to ensure transparency.
Comment 10: It is important to justify why the selected experts are considered representative of all significant stakeholders in this area. Providing details on their demographics, locations, and the timeline and location of the four meetings would strengthen the study's credibility.
Comment 11: The process of expert validation should be described in greater detail, including how experts were chosen and the method of feedback collection.
Comment 12: Clarify whether each expert read the articles included in the systematic review and explain the connections between the systematic review and the desk study.
Results
Comment 13: The synthesis of findings from the systematic review is somewhat superficial. A more detailed comparison of study characteristics and outcomes would enhance this section.
Comment 14: The results of the desk study could benefit from a more detailed explanation of how the conceptual framework was developed and refined based on the literature and expert feedback.
Discussion
Comment 15: The discussion lacks a comparison with findings from previous studies. Including such comparisons would help contextualize the results within the broader field of research.
Comment 16: The conclusion could be more specific about the contributions of the study to the existing body of knowledge. The theoretical and practical implications should be better addressed to highlight the study's impact.
Comment 17: The discussion could be more critical, addressing potential biases in the systematic review and desk study. This would provide a more balanced view of the study's findings and limitations.
References
Comment 18: There should be a clearer differentiation between references used in the systematic review and those used in the desk study. This distinction would enhance the clarity and organization of the references section.
By addressing these points, the paper could significantly improve its clarity, transparency, and overall contribution to the field. Thank you again for the opportunity to review this important work.
Comments on the Quality of English LanguageThe overall use of English in this research paper is clear and generally effective in conveying the intended messages. However, there are areas where the language could be polished to enhance readability and precision. The paper occasionally exhibits awkward phrasing and grammatical errors that may detract from the clarity of the arguments presented.
Author Response
Comment 1: [The data and citations used in the introduction are outdated. Updating the introduction with more recent references would help set the context more clearly and reflect the latest research in this field.]
Response 1: [Citations were updated to match with the period covered by the systematic review]
Comment 2: [The introduction dedicates too much space to general information about COVID-19. This section could be shortened, and more space allocated to reviewing literature directly related to the research question.]
Response 2: [Well addressed this suggestion and removed some sections from the introduction]
Comment 3: [In lines 121-122, the paper mentions “a fully integrated system for epidemic and pandemic preparedness and response in a multi-hazard context.” This statement should be supported by reviewing additional research and previous frameworks to provide a stronger foundation.]
Response 3: [fairly addressed this in the introduction]
Comment 4: [The introduction could be more specific in outlining the exact gaps the study intends to fill. Clearly articulating the rationale behind the research questions would provide better context for the study.]
Response 4: [addressed this, and gap was highlighted in the introduction (lines 71-88)]
Comment 5: [The search terms and strategies are not detailed enough. Including the date range, search terms, and any Boolean operators used would enhance the transparency and replicability of the review.]
Response 5: [Well addressed ]
Comment 6: [The exclusion criteria need more detail. Providing a clearer rationale for excluding certain studies would improve the methodology's robustness.]
Response 6: [Exclusion criteria were further elaborated]
Comment 7: [Lines 186-187 mention that six grey literature sources were included in the literature review. The criteria for selecting these sources should be clearly explained to justify their inclusion.]
Response 7: [addressed and added a new sentence (173-175)]
Comment 8: [The paper should detail the exact variables extracted from each study and the process used to develop the framework or variables during the systematic review stage. This would enhance the study's transparency and replicability.]
Response 8: [As a strategy, manuscripts were properly cited in each component, which developed based on the systematic review to reflect variables of manuscripts selected]
Comment 9: [The methodology for the desk study should be more detailed. The criteria for selecting documents and reports should be clearly outlined to ensure transparency.]
Response 9: [This is well addressed in lines 101-106]
Comment 10: [It is important to justify why the selected experts are considered representative of all significant stakeholders in this area. Providing details on their demographics, locations, and the timeline and location of the four meetings would strengthen the study's credibility.]
Response 10: [A new section was added to justify this (see lines 227-238)
Comment 11: [The process of expert validation should be described in greater detail, including how experts were chosen and the method of feedback collection.]
Response 11: [the section of the validation process was substantially revised to address this aspect]
Comment 12: [Clarify whether each expert read the articles included in the systematic review and explain the connections between the systematic review and the desk study.]
Response 12: [A sentence added to clarify this (see lines 761-763)
Comment 13: [The synthesis of findings from the systematic review is somewhat superficial. A more detailed comparison of study characteristics and outcomes would enhance this section.]
Response 13:[Fairly address this aspect during the revisions]
Comment 14: [The results of the desk study could benefit from a more detailed explanation of how the conceptual framework was developed and refined based on the literature and expert feedback.]
Response 14: [revised the section according to this suggestion]
Comment 15: [The discussion lacks a comparison with findings from previous studies. Including such comparisons would help contextualize the results within the broader field of research]
Response 16: [this suggestion was substantially addressed by adding a section (see lines 630-643)
Comment 16: [The conclusion could be more specific about the contributions of the study to the existing body of knowledge. The theoretical and practical implications should be better addressed to highlight the study's impact.]
Response 16: [Conclusion was revised addressing this suggestion)
Comment 17: [The discussion could be more critical, addressing potential biases in the systematic review and desk study. This would provide a more balanced view of the study's findings and limitations.]
Response 17: [Well addressed this suggestion and a section included in the discussion (see lines 822-837)
Round 2
Reviewer 2 Report
Comments and Suggestions for Authors
The authors responded adequately to my comments on the original version by providing more details on the criteria for selection of experts and the validation process, in addition to improving English language.
Author Response
Comments: The authors responded adequately to my comments on the original version by providing more details on the criteria for selection of experts and the validation process, in addition to improving English language.
Response: We really appreciate your insightful comments and suggestions
Reviewer 3 Report
Comments and Suggestions for Authors
I appreciate the author’s efforts to revise the manuscript in response to the feedback provided. The updates, including the enhancement of citations and the refinement of the introduction and discussion, have significantly strengthened the work. However, to further improve the manuscript, I recommend focusing on adding more content that reviews literature directly related to the research question. While the background information has been streamlined, it is crucial to provide a more detailed review of recent and relevant studies pertinent to the research question. This addition will offer a clearer and more focused context for the study, helping to better frame its contributions and ensure that the introduction addresses the specific gaps identified in the literature.
Author Response
Comments 1: However, to further improve the manuscript, I recommend focusing on adding more content that reviews literature directly related to the research question. While the background information has been streamlined, it is crucial to provide a more detailed review of recent and relevant studies pertinent to the research question.
Response 1: We addressed this very important suggestion in the introduction section using recent research works. (see the lines 102-124 particularly)